# Leucine-Enriched Protein Supplementation Increases Lean Body Mass in Healthy Korean Adults Aged 50 Years and Older: A Randomized, Double-Blind, Placebo-Controlled Trial

**DOI:** 10.3390/nu12061816

**Published:** 2020-06-18

**Authors:** Yeji Kang, Namhee Kim, Yong Jun Choi, Yunhwan Lee, Jihye Yun, Seok Jun Park, Hyoung Su Park, Yoon-Sok Chung, Yoo Kyoung Park

**Affiliations:** 1Department of Medical Nutrition, Kyung Hee University, Yong-in 17104, Korea; yejikang@khu.ac.kr (Y.K.); namheekim@khu.ac.kr (N.K.); 2Department of Endocrinology and Metabolism, Ajou University School of Medicine, Suwon 16499, Korea; colsmile@ajou.ac.kr; 3Department of Preventive Medicine and Public Health, Ajou University School of Medicine, Suwon 16499, Korea; yhlee@ajou.ac.kr (Y.L.); dream10307@naver.com (J.Y.); 4Health & Nutrition R&D Group, Maeil Dairies Co., Ltd., Pyeongtaek-si 17714, Korea; sj.park@maeil.com (S.J.P.); parkhs@maeil.com (H.S.P.)

**Keywords:** lean body mass, sarcopenia, leucine, protein

## Abstract

Early prevention of sarcopenia could be an important strategy for muscle retention, but most studies have focused on subjects aged 65 or older. Therefore, in this study we investigated the effects of leucine-enriched protein supplementation on muscle condition in a sample including late middle-aged adults. A 12-week intervention was performed for 120 healthy community-dwelling adults by providing either leucine-enriched protein supplement [protein 20g(casein 50%+ whey 40%+ soy 10%, total leucine 3000 mg), vitamin D 800IU(20 ug), calcium 300 mg, fat 1.1 g, carbohydrate 2.5 g] or isocaloric carbohydrate supplement twice per day. Appendicular skeletal muscle mass (ASM) and lean body mass (LBM) were measured by dual-energy X-ray absorptiometry. A total of 111 participants completed the study, with a dropout rate of 9.2%. LBM normalized by body weight (LBM/Wt) was significantly increased (*p* < 0.001) in the intervention group (0 wk: 63.38 ± 0.85 vs. 12 wk 63.68 ± 0.83 in the intervention group; 0 wk: 63.85 ± 0.82 vs. 12 wk: 63.29 ± 0.81 in the control group). In subgroup analyses, significant differences remained only in subjects between 50 and 64 years of age. We concluded that leucine-enriched protein supplementation can have beneficial effects by preventing muscle loss, mainly for late middle-aged adults.

## 1. Introduction

Healthy aging, along with healthy living, is becoming more important due to the extension of life expectancy. According to data from Health Statistics Korea, the life expectancy of Koreans averaged 82.7 years in 2017, and the elderly aged 65 and older accounted for 15.5% of the population in 2019 [1]. As the elderly population increases, interest in maintaining health in old age is also increasing. The preservation of skeletal muscle mass is necessary for maintaining physical function and the muscle strength necessary for physical activity [2]. Sarcopenia, which increases with age, increases the risk of falls and fractures due to decreases in athletic ability. In addition, loss of independence due to decrease in activities of daily living (ADL) increases the risk of death [3]. When sarcopenia occurs, the risk of physical disability in daily life increases four-fold, and the risk of falls increases two- to three-fold when balance is impaired and the use of braces such as walking disorders and canes increases [2].

Sarcopenia is also strongly associated with decreased protein intake. Houston et al. (2009) reported that 40% of elderly people over 70 years of age do not meet the recommended protein intake, and a significant increase in sarcopenia was reported in the elderly who consumed less than the recommended amount [4]. Moreover, even in Korean adults over 50, the prevalence of skeletal muscle mass reduction was significantly higher in the group that did not meet the recommended protein intake suggested by the Korean Nutrition Society [5]. Protein intake is necessary for stimulating muscle protein assimilation, stimulating systemic protein synthesis, and preventing excessive muscle protein degradation. If protein intake is increased to 0.8 g/kg/day, muscle synthesis is increased, and muscle strength is improved [6]. It is also important to monitor the efficacy of branched-chain amino acids (BCAA) that have better effects on muscles. Leucine, which plays a key role in muscle health, stimulates protein synthesis through the activation of intracellular signaling pathways that accelerate the initiation of mRNA translation [7]. It is also noteworthy that leucine mediates more immediate control of protein synthesis and includes regulatory steps that respond rapidly to metabolic disturbances of amino acid utilization [8]. Previous studies show that when leucine-added protein was supplied to sarcopenic elderly people for 12 weeks, muscle mass in the limbs and total lean mass were simultaneously increased. When 6 g/day of leucine was provided to institutionalized elderly people for 13 weeks, functional performance was improved [9,10]. However, previous studies of protein supplementation interventions usually focused on elderly people over 65 years of age, and research on the pre-elderly is rare [6,9,11,12,13]. For example, a recently published nutritional intervention study examined the effects of providing 1.1 g or 1.3 g/kg/day of leucine for 12 weeks in a sample including only frail elderly people over 65 years of age [14]. Another study found that short physical performance battery (SPPB) level and lean body mass improved after providing 1.0 g/kg/day leucine, but was also limited to elderly individuals over 65 [15]. Micronutrients such as calcium and vitamin D also play important roles in muscle health. Calcium is involved in muscle contraction and relaxation [16], and vitamin D promotes muscle synthesis as well as calcium absorption, thereby promoting protein synthesis by increasing the concentration of ATP inside the cell [17]. Vitamin D, particularly, plays an important role in skeletal muscle. The deficiency of vitamin D at the tissue level is associated with proximal myopathy, muscle fiber atropy, which can be reversed by administration of vitamin D [18].

Therefore, this study hypothesized that continuous intake of protein supplements, including leucine, that have a special effect on muscle synthesis, would be effective for absorption by the body and increase muscle mass, muscle strength, and physical function independent of the increase in physical activity. The effects of 12 weeks consumption of a leucine-enriched protein, calcium, and vitamin D micronutrient mixture on muscle health status in non-sarcopenic Korean adults aged 50 to 64, as well as older adults aged 65 to 80, were investigated.

## 2. Materials and Methods

### 2.1. Study Population

From July 2019 to January 2020, healthy adults aged 50 to 80 years were recruited and the study was conducted at Ajou University Hospital, Gyunggi-do, Korea. All subjects gave their informed consent for inclusion before they participated in the study. The study was conducted in accordance with the Declaration of Helsinki, and the protocol was approved by the Ethics Committees of the Institutional Review Boards (IRBs) of both Kyung Hee University and Ajou University Hospital in April 2019 (No. KHSIRB 2019-004, No. AJIRB-MED-FOD-1948). In addition, this study was registered at the Clinical Research Information Service (No. KCT0005111), which is a non-profit online registration system for clinical trials to be conducted in Korea established by the Korea Centers for Disease Control and Prevention (KCDC). The KCDC has joined the WHO International Clinical Trials Registry Platform (ICTRP) as the 11th member of its Primary Registry, and the full trial protocol for this study is avaliable here.

The inclusion criteria for this study were age over 50 and under 80 years, understanding the purpose of this study, voluntarily agreeing to participate in the study, body mass index (BMI) of 18.5 to 27 kg/m², appendicular skeletal muscle mass index (ASMI) over 7.0 kg/m² for men and 5.4 kg/m² for women as measured by dual energy X-ray absorptiometry (DXA; GE Healthcare, Madison, WI, USA). The exclusion criteria were (1) eGFR 60 mL/min/1.73 m^2^ or less, (2) currently receiving insulin therapy or taking steroids and testosterone-based drugs, (3) having chronic lung disease, (4) taking vitamin D 1000 IU or more per day, (5) consuming a special diet for the purpose of disease management, (6) history of high-intensity strength training in the past 6 months, (7) musculoskeletal disorders that make it difficult to exercise, (8) untreated or unregulated cardiovascular disease that may affect muscle mass or exercise, (9) period of less than 5 years after the treatment of malignant tumors, and (10) presence of cirrhosis, diabetes, and other chronic diseases. Initially, 156 participants were screened to participate in this study, but 36 participants were excluded because they did not meet the selection criteria, primarily due to the results of DXA. When the number of participants reached 120, we terminated recruiting.

### 2.2. Sample Size Determination

A priori analysis determined that a sample size of 102 would be needed to detect a least significant change of 0.151 (SD = 0.301) in ASM with DXA. This would achieve 90% power to detect significant difference at 0.05 alpha level. Considering a dropout rate of 15% we aimed to recruit 120 volunteers.

### 2.3. Intervention and Random Assignment Blinding

The participants were assigned to consume either protein mixture powder (Muscle Health Solution Formula, Maeil Dairies Co.,Ltd., Gyeonggi-do, Korea), [protein 20 g(casein 50%+ whey 40%+ soy 10%, total leucine 3000 mg), vitamin D 800IU(20 ug), calcium 300 mg, fat 1.1 g, carbohydrate 2.5 g] or isocaloric-placebo supplement powder [carbohydrate 25 g] twice a day for 12 weeks. Nutritional interventions for each group included daily consumption of the powder supplement or placebo powder dissolved in water or milk at least twice, which was not controlled in respect of the preferences of the subjects. Dietary intake was measured by 24-h recall tasks administered by a trained dietitian before and after the study. Subjects were all instructed to do only a light exercise sequence depending on their performance level. Exercise monitoring was conducted by providing a 20 min resistance-exercise sequence brochure designed by a registered health trainer. In addition, International Physical Activity Questionnaires (IPAQ) were conducted at the beginning of the study and after the study, to monitor changes in physical activity.

All participants were assigned a number according to registration order before conducting the trial. A block randomization consisting of block sizes of four, including interventions (A) and control groups (B) at a 1:1 ratio, was prepared in advance from a random number sequence generated using SAS 9.4 by a researcher who was not involved in the registering of the participants. Participants were assigned to the intervention or control group according to the order of assignment. All other researchers involved in the subject’s intervention, data collection, analysis, and statistical analysis, except those developing the product, were blinded to treatment allocation until the statistical analysis was completed.

### 2.4. Muscle Health

#### 2.4.1. Primary Outcomes: Muscle Mass

Appendicular skeletal muscle mass index (ASMI) (kg/m^2^) = ASM (kg)/Height (m^2^), ASM normalized by body weight (ASM/Weight,%) = ASM (kg)/body weight (kg) × 100, ASM normalized by BMI (ASM/BMI) = ASM (kg)/BMI (kg/m^2^) and lean body mass (kg) normalized by height (LBM/Height^2^) = LBM (kg)/Height (m^2^), lean body mass (kg) normalized by body weight (LBM/Weight, %) = LBM (kg)/body weight (kg) × 100, lean body mass normalized by BMI (LBM/BMI) = LBM (kg)/BMI (kg/m^2^) was assessed by using DXA after at least eight hours of fasting. Height was measured using an automatic height and weight measuring device (GL-150; G-tech International, Uijeongbu, Korea), while arm and calf circumference were measured using a tape measure.

#### 2.4.2. Secondary Outcomes: Muscle Strength

Muscle strength was measured before and after the study. For the measurement of femoral muscle strength, a Commander Echo Muscle Tester (Micro FET2, HOGGAN Ergo, USA) was used to measure the right and left legs three times each, with the average values being recorded. In addition, the grip strength (kg) of the right and left hands were each assessed three times using a dynamometer (Jamar, Plus+ Digital Hand Dynamometer, Preferred, USA) and the average values and standard deviations were recorded.

#### 2.4.3. Physical Performance

The short physical performance battery (SPPB) test was used to measure physical performance. An evaluation table was used to measure and record standing balance, 4 m gait speed, and repeated chair stands. For each item, 0 points equated to inability to perform and 1 to 4 points were assigned depending on task performance. Finally, a score of 12 points indicated that the subject had gotten 4 points for each task. All testing was carried out by a single expert inspector. The physical performance of each subject was assessed at the beginning and at the end of the study.

### 2.5. Blood Analysis

Blood analyses were performed before and after the study. Each participant’s blood was collected after at least eight hours of fasting. Blood samples were drawn by a registered phlebotomist and CBC (Complete blood cell count), albumin, blood sugar, insulin, ALT, AST, creatinine, eGFR, and 25(OH)D were measured in the hospital laboratory.

### 2.6. Statistical Analysis

The data were analyzed with the Statistical Package for the Social Sciences (SPSS) version 25.0 (IBM, Seoul, Korea). To see if there was a difference in baseline between the two groups, an independent T test was performed for the continuous variable and a chi-square test for the categorical variable. Changes between baseline and after 12 weeks were analyzed using a Paired *t*-test, and general characteristics of the subjects were analyzed using descriptive statistics and expressed as *n* (%). Using SAS version 9.4 (SAS Institute Inc., Cary, NC, USA) to see if there was an intervention effect after 12 weeks, analysis was conduted using a mixed effect model repeat measurement (MMRM). The MMRM included the fixed effect of time, group, and time × group and was analyzed by correcting the age difference in baseline between the two groups. Additionally, only participants who completed all follow-up studies (*n* = 111) were analyzed (per protocol analysis, PP). *p* values < 0.05 were considered statistically significant.

## 3. Results

### 3.1. Baseline Characteristics of Participants

There was no significant difference (*p* = 0.307) in the proportions of males and females in the intervention and control groups. Data for body composition, muscle function, blood test and, CBC are shown in Table 1 and did not significantly differ between the intervention and control groups. However, the age of the participants was significantly different between groups (61.23 ± 6.87 years for the intervention group and 58.38 ± 5.72 years for the control group, *p* = 0.015). eGRF data are not shown because all participants had levels greater than 60 mL/min/1.73 m^2^ both before and after the study.

### 3.2. Dropouts and Compliance

A total of 111 datasets from 54 participants in the intervention group and 57 controls were used in the final analysis, which included participants for whom follow-up was completed (Figure 1). During the study, a total of nine participants dropped out (dropout rate of 9.2 percent). Six participants were dropped from the intervention group, due to enrolling error (*n* = 1), withdrawal of consent (*n* = 4), and loss of follow-up (*n* = 1). In the control group, three participants were dropped for reasons such as non-compliance (*n* = 2) and loss of follow-up (*n* = 1). Reasons for withdrawal of consent were adverse reactions such as constipation (*n* = 1), heartburn (*n* = 1), and skin rash (*n* = 1), as well as simple change of mind (*n* = 1). To assess compliance, the participants were asked to record the number of supplements they consumed every day, and reported compliance of more than 80%.

### 3.3. Dietary Intake

Energy intake in both groups did not differ based on self-reported habitual intake. Intakes of vitamin D significantly increased from baseline to 12 weeks in the control group but decreased in the intervention group (from 2.467 ± 0.40 to 4.317 ± 0.67 µg in the control group vs. from 3.260 ± 0.42 to 2.904 ± 0.68 µg in the intervention group) (*p* value for time × group interaction = 0.047). Riboflavin was significantly increased in the control group (from 1.208 ± 0.06 to 1.552 ± 0.08 mg; *p* value for time × group interaction = 0.047), but significantly decreased in the intervention group (from 1.236 ± 0.06 to 1.218 ± 0.08 mg; *p* value for time × group interaction = 0.017). No significant effects were observed for any other nutrients (Table 2).

### 3.4. Effects of Intervention on Muscle Health and Physical Performance

After the 12-week intervention, there was a significant time effect of ASMI kg/m^2^ between groups, which increased in both groups (p value for time effect = 0.032). Lean body mass/height significantly increased in the intervention group compared to the control group (from 15.06 ± 0.21 to 15.29 ± 0.21 vs. from 15.04 ± 0.21 to 15.09 ± 0.22) (p value for time × group interaction = 0.019). Lean body mass/weight significantly increased in the intervention group compared to the control group at week 12 (from 63.38 ± 0.85 to 63.68 ± 0.83 vs. from 63.85 ± 0.82 to 63.29 ± 0.81) (p value for time × group interaction <0.001). Lean body mass/BMI was significantly increased in the intervention group compared to the control group at week 12 (from 1.64 ± 0.03 to 1.65 ± 0.03 vs. from 1.65 ± 0.03 to 1.63 ± 0.03) (p value for time × group interaction = 0.001). Arm circumference exhibited a significant time effect between groups, by 0.37 ± 0.10 and 0.32 ± 0.00 cm in the intervention and control groups, respectively. Although there were significant time effects on BMI (kg/m^2^), weight (kg), and arm circumference (cm) (p < 0.001), no significant effects were observed on calf circumference (cm). No significant effects were observed on femoral muscle strength (from 182.35 ± 7.08 to 182.70 ± 5.86 N in the intervention group and from 173.91 ± 6.89 to 172.39 ± 5.70 N in the control group), femoral muscle strength/weight (from 2.98 ± 0.11 to 2.95 ± 0.09 N/kg in the intervention group and from 2.85 ± 0.11 to 2.83 ± 0.09 N/kg in the control group), or grip strength (from 28.99 ± 1.04 to 28.51 ± 1.05 kg in the intervention group and from 26.53 ± 1.02 to 25.89 ± 1.02 kg in the control group). After the 12-week intervention, we observed a significant time effect only on the SPPB score in both groups (Table 3).

### 3.5. Effects of Intervention on Blood Measurements

At 12 weeks, insulin levels remained within reference values but had increased from baseline in the control group and decreased significantly in the intervention group (from 6.54 ± 0.43 to 7.97 ± 0.48 uIU/mL in the control group vs. from 7.51 ± 0.44 to 7.30 ± 0.49 uIU/mL in the intervention group) (*p* value for time × group interaction = 0.019). However, 25(OH)D decreased from baseline to 12 weeks in the control group while it increased significantly in the intervention group (from 35.30 ± 1.44(sufficiency) to 29.24 ± 1.36 µg/mL (insufficiency) in the control group vs. from 34.40 ± 1.44(sufficiency) to 39.83 ± 1.53 µg/mL (sufficiency) in the intervention group) (*p* value for time × group interaction < 0.001). Glucose, ALT, AST, creatinine, and CBC did not significantly change over time in either group (Table 4).

### 3.6. Subgroup Analysis: By Age

Further analyses were performed on muscle health effects according to the age of the participants (50–64 years vs. 65 years and older) (Table 5 and Table 6). Lean body mass/height increased significantly in both groups (from 14.942 ± 0.22 to 14.943 ± 0.22 in the control group vs. from 15.091 ± 0.25 to 15.277 ± 0.24 in the intervention group; *p* value for time × group interaction = 0.033), with higher improvement in the intervention group compared with the control group. Lean body mass/weight significantly increased in the intervention group only (from 64.03 ± 1.01 to 64.37 ± 0.98 vs. from 64.09 ± 0.90 to 63.45 ± 0.88; *p* value for time × group interaction < 0.001). Lean body mass/BMI increased with a significant time × group interaction in the intervention group (*p* = 0.001). However, in participants aged 65 years and older, there were no significant differences in baseline characteristics between intervention groups.

## 4. Discussion

In this study, we explored whether the provision of a leucine-enriched protein mixture would help individuals over the age of 50 to maintain or strengthen their muscle health when muscle reduction began to occur, and/or would help maintain muscle mass and function among elderly individuals for whom strength and muscle function reduction had already occurred. Twelve weeks of nutritional intervention did not lead to significant improvements in SPPB or muscle strength. However, there was improvement in muscle mass in the intervention group compared to the control group, which suggests that a complex of leucine-enriched protein, calcium, and vitamin D, which has important effects on muscle growth, may prevent and ameliorate sarcopenia.

This area of research primarily focuses on the acute impact of muscle protein synthesis on frail elderly individuals who have already experienced significant muscle loss. Only a few long-term nutritional supplemental intervention studies have been conducted to evaluate changes preventing sarcopenia among healthy middle-aged and older adults. In recent studies of pre-frail and frail people (aged 65 and older), Tieland et al. reported improvements in SPPB when subjects were supplied with nutritional supplements, but supplementation did not result in differences in muscle mass or strength [19]. Another recent study by Park et al. showed improvements in ASM and gait speed in subjects who received 1.5 g protein/kg/d, but did not observe differences in muscle mass or physical performance between groups supplemented with 1.2 and 0.8 g protein/kg/day [13]. In a recent study that tested the effects of leucine alone, the effect of leucine supplementation on sarcopenia for 13 weeks resulted in improved functional performance, lean mass index, and maximum static expiratory force. However, this study was performed among a sample living in a nursing home (Valencia, Spain) who are over 65 years of age [10].

Leucine, a nutritional supplement that can positively affect muscle mass and increase muscle strength, is an amino acid (protein building block) that is most often stored in the body in the form of branched chain amino acids with isoleucine and valine [20]. Leucine stimulates the mTOR (mammalian target of rapamycin) signaling pathway and is known to positively affect muscle mass by increasing muscle protein synthesis and reducing muscle protein degradation [21,22]. In addition, leucine can stimulate insulin release by pancreatic beta cells, and thus, its beneficial effects may improve glucose in skeletal muscle, increase metabolic signals, and positively contribute to maintenance of muscle mass [23]. In an intervention study that temporarily supplied participants with whey protein and leucine (20 g whey protein, 3 g total leucine, 9 g carbohydrates, 3 g fat, 800 IU (20 µg) vitamin D, and a mixture of vitamins, minerals, and fibers), researchers observed high levels of essential amino acids, especially leucine. Leucine is necessary to elicit acute muscle protein synthesis response [9]. In addition, the habitual consumption of high levels of protein and essential amino acids is related to the maintenance of lean body mass [24,25]. Similar to the above mentioned studies, when subjects were categorized into three stage groups according to sarcopenia, interventional studies with whey protein and leucine-enriched supplements resulted in high lean body mass, muscle quality, grip strength, and gait speed in the severe sarcopenia group [9].

In this study, however, we included healthy older people with higher ASMI, and this ASMI was used as the primary effective indicator of a change in muscle mass from DXA results (ASMI, lean body mass). ASMI was increased by about 0.05 kg/m^2^ in the control group, and by about 0.07 kg/m^2^ in the intervention group. Lean body mass/height increased in both groups, but lean body mass/weight and lean body mass/BMI increased only in the intervention group. On the other hand, ASMI, unlike LBM, did not show any statistical difference. We speculate that the differences in these observations could be explained by the composition accounted for each parameter, While LBM includes trunk area (muscle) with bone tissues which represents a relatively more stable value, ASMI reflects only the muscle mass from the four limbs [26]. The secondary effective indicator in this study was muscle strength and physical performance (femoral muscle strength knee extension, grip strength, SPPB), but there was no significant effect associated with interventions in femoral muscle strength knee extension and grip strength. Only the level of SPPB showed a positive time effect for both groups. Unlike our expectation, and similar to the muscle strength result, protein supplementation did not improve the muscle function measured by SPPB. This can be explained by the fact that the average score of SPPB before and after the study was all within the healthy, normal scale provided by the National Institute on Aging (NIA), as a total score of 11 points or more [27].

The level of daily physical activity assessed by IPAQ did not change throughout this study in either group, supporting the hypothesis that muscle mass was not affected by physical activity. Similar results in previous studies that measured muscle, muscle strength, and physical performance showed that in elderly people observed for 12 weeks, muscle mass (ASM, SMI) increased, and walking speed improved [13]. In addition, healthy older males who consumed a medical nutrition drink (21 g whey protein, 3 g leucine, 3 g fat, and vitamin D 800IU (20 µg), 500 mg of calcium) before breakfast for 6 weeks experienced greater improvements in muscle mass (appendicular lean mass, leg lean mass) than the control group, while grip strength and SPPB did not differ between groups [28].

Exercise is considered the standard treatment to increase muscle mass and improve physical performance in adults with sarcopenia. However, in this study, we aimed to investigate the sole effects of nutritional intervention to exclude the effects of exercise. Our results suggest that functional performance does not necessarily parallel muscle mass change. Serum 25-hydroxy vitamin D concentrations significantly increased to 40 ng/mL after 12 weeks in the intervention group, who consumed 800IU (20 µg) of vitamin D twice daily through supplements and exhibited high compliance with nutritional supplement intake. A serum 25-hydroxy vitamin D concentration of 24–30 ng/mL is suggested to be optimal for the prevention of falls and fractures [29], according to a study in which vitamin D 800IU (20 µg)/d was supplied to active participants for more than 3 months, resulting in vitamin D concentrations within the optimal range. Vitamin D supplementation in combination with leucine and protein could have contributed to the positive effects observed on muscle parameters in this study [18]. Unexpectedly, elevated levels of insulin were observed in the control group, which could be partially explained by taking a total of 50 g of carbohydrates daily from the isocaloric supplement. Although the level has significantly increased from the baseline level, the increased level of insulin remained within the reference values.

The dietary intake analysis presented in Table 2 included only meal intake, excluding nutrients from supplementation. The authors did not intentionally include the nutritional value of supplements in the dietary intake analysis, in order to strictly separate the supplement effect from the changes in the dietary intake. At baseline, the daily protein intake of the two groups exceeded the recommended daily intake for Korean adults, 0.91 g/kg. Considering that the participants were provided either high-protein supplements or isocaloric protein-free supplements, the total protein intakes of the intervention group reached a high of ~1.50 g/kg per day (vs. in the control group after 12 weeks = 1.05 ± 0.35 g/kg/day), as recommended by the European Society for Clinical Nutrition and Metabolism for geriatric patients and PROT-AGE (1.2–1.5 g/kg/day) [30]. Interestingly, intake of vitamin D and riboflavin was increased only in the control group. This could be attributed to the higher consumption of protein-rich drinks such as milk and soy milk, in which the isocaloric-placebo mixture was dissolved. This was not the case in the intervention group, in which the participants drank more water instead of milk or soy milk. Adults over 40 can lose about 1 percent of muscle mass per year, and older adults can lose up to 5 percent per year, due to age-related decreases in growth hormones. In addition, the maximum isometric contraction is reduced by about 20% by age 50, and the reduction by age 70 is about 50% [31]. In general, 25% of elderly aged 70 or younger and about 40% of elderly aged 80 or older show symptoms of sarcopenia [31,32]. Skeletal muscles are highly adaptable to stimuli and react quickly to nutritional or mechanical loads to grow [33,34].

When we analyzed data by age, there was a significant increase inlean body mass/height in both intervention and control groups for participants 50–64 years of age, but the increase was greater in the intervention group. This is in line with the results of previous studies that concluded that providing moderate amounts of protein effectively stimulates muscle protein synthesis in healthy adults [12]. On the other hand, participants 65 years or older experienced no significant changes in muscle strength, muscle mass, or physical performance. However, previous studies usually showed improvements of muscle mass by supplying high-protein nutrient supplements to frail elderly people [19], while numerous studies showed improvements in muscle mass and protein synthesis by supplying high protein supplements to elderly people over 65 years of age [35,36,37]. The discrepancy between the results of previous studies and the current study lies in the health status of the participants. We included subjects who showed no sarcopenic indicators and who did not meet the definition of ‘frail’. In addition, previous studies have reported that healthy adults over the age of 40 lose about 8% of their muscle mass every 10 years, and healthy adults lose an average of 24% of their muscle between the ages of 40 and 70 years [38,39]. In this regard, muscle health outcomes in older adults over age 65 years of age in this study may be due to a decrease in natural muscle mass. Therefore, in this study, comparative analysis was performed through the control group. As a result, no significant change in muscle mass was observed in the control group. Moreover, according to previous studies, researchers are trying to explain sex differences in relation to protein metabolism [40,41,42]. The results of protein metabolism during resistance exercise in older women showed less increase in muscle protein synthesis compared to men [40,43]. Since the proportion of women in the 65-years-and-older population was high, we may not have gained significant results for muscle health.

There are some limitations to this study. First, the participants may have become accustomed to testing at baseline, and familiarity with the tests could have affected scores when testing strength and physical performance after 12 weeks. However, these effects were not biased to specific groups and can only explain the time effect observed. Second, we did not strictly instruct the subjects on which kind of drink the supplements should be dissolved into and we let the subjects consume the supplement at any time of the day. A previous study [28] showed that leucine-enriched protein nutritional supplement intake time is an important factor in improving muscle mass and strength. Third, in this study, no specific exercise program was planned in addition to the distribution of exercise books provided to all participants. Fourth, it was difficult to recruit healthy adults who met the inclusion criteria of this study, and for this reason, the number of participants was not evenly matched by age and sex. Fifth, the intervention period of this study was only 12 weeks, which was relatively short compared to previous studies.

Despite these limitations, this study has several advantages. The strict exclusion criteria (including DXA results) were used to accurately reflect the population of healthy aging people. We monitored and limited further increases in exercise in order to isolate the effect of the leucine-enriched protein supplement. Lastly, dietary intake was monitored and participants were watched closely by a trained dietitian throughout the study period in order to make sure the participants met the goal of ingesting a daily protein intake of ~1.50 g/kg of protein per day in the intervention and control groups.

## 5. Conclusions

In conclusion, in this study we investigated the effects of leucine-enhanced protein supplementation for 12 weeks, which specifically affects muscle synthesis, in pre-elderly and elderly subjects. Our subjects maintained high intake compliance, and supplementation was effective for increasing muscle mass, and specifically more effective in subjects aged 50–65. We conclude that early prevention of sarcopenia may be an important strategy for muscle retention. Prevention and early intervention activities should start earlier in life rather than after the onset of sarcopenia.

## Figures and Tables

**Figure 1 nutrients-12-01816-f001:**
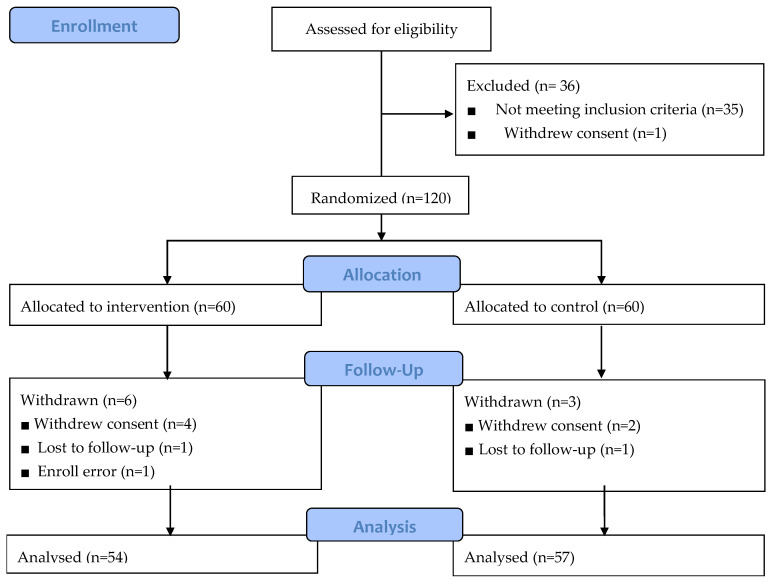
Flow chart of the study population.

**Table 1 nutrients-12-01816-t001:** Baseline characteristics of subjects.

	Control (*n* = 60)	Intervention (*n* = 60)	*p*
Age	58.38 ± 5.72	61.23 ± 6.87	0.015 *
**Sex**			0.307
Men (%)	14 (23.33)	19 (31.67)	
Women (%)	46 (76.67)	41 (68.33)	
Weight (kg)	60.74 ± 6.41	61.17 ± 8.44	0.759
Height (cm)	160.4 ± 6.21	160.2 ± 7.50	0.905
BMI (kg/m^2^)	23.59 ± 1.84	23.74 ± 2.06	0.678
**Muscle mass**			
ASM (kg)	16.70 ± 3.13	17.07 ± 3.84	0.564
ASMI (kg/m^2^)	6.45 ± 0.83	6.85 ± 0.96	0.434
ASM/Wt (kg/kg,%)	27.39 ± 3.33	27.71 ± 3.27	0.597
ASM/BMI [kg/(kg/m^2^)]	0.71 ± 0.13	0.71 ± 0.14	0.735
LBM (kg)	38.76 ± 6.30	39.12 ± 7.25	0.766
LBM/Ht(kg/m^2^)	14.98 ± 1.58	15.11±1.61	0.663
LBM/Wt(kg/kg,%)	63.69 ± 6.50	63.81 ± 5.98	0.916
LBM/BMI [kg/(kg/m^2^)]	1.64 ± 0.26	1.65 ± 0.27	0.971
**Muscle function**			
Femoral muscle strength (N)	171.7 ± 55.57	183.0 ± 45.35	0.223
Femoral muscle strength/Wt (N/kg)	2.82 ± 0.84	3.02 ± 0.78	0.178
Grip strength (kg)	26.37 ± 6.98	28.92 ± 7.93	0.065
SPPB (score)	11.53 ± 0.83	11.63 ± 0.66	0.468
Arm circumference (cm)	29.86 ± 1.75	30.32 ± 2.17	0.199
Calf circumference (cm)	34.75 ± 2.24	35.07 ± 2.84	0.495
**IPAQ**	2120.0 ± 1770.4	2174.2 ± 2158.1	0.881
**Blood test**			
Albumin (g/dL)	4.60 ± 0.20	4.56 ± 0.24	0.256
Glucose (mg/dL)	97.70 ± 6.29	99.60 ± 9.28	0.192
Insulin (uIU/mL)	6.56 ± 2.11	7.44 ± 3.91	0.130
ALT (GPT) (U/L)	19.33 ± 8.47	21.40 ± 10.60	0.241
AST (GOT) (U/L)	22.53 ± 6.29	24.33 ± 8.44	0.188
Creatinine (mg/dL)	0.75 ± 0.13	0.78 ± 0.15	0.355
25(OH)D (ng/mL)	34.99 ± 11.10	34.71 ± 11.20	0.892
**CBC**			
WBC (×10^3^/μL)	5.10 ± 0.91	5.37 ± 1.34	0.218
RBC (×10^6^/μL)	4.33 ± 0.29	4.42 ± 0.41	0.167
Hb (g/dL)	13.41 ± 1.17	13.79 ± 1.17	0.077
HCT (%)	40.30 ± 3.11	41.41 ± 3.29	0.059
MCV (fL)	93.02 ± 4.99	93.75 ± 4.11	0.388
MCH (pg)	30.97 ± 2.09	30.76 ± 3.86	0.705
MCHC (g/dL)	33.26 ± 0.77	33.25 ± 0.62	0.917
RDW (%)	13.30 ± 1.18	13.19 ± 0.54	0.541
PLT (×10^3^/μL)	227.3 ± 51.85	214.6 ± 42.23	0.144
MPV (fL)	7.73 ± 0.77	7.82 ± 0.70	0.492

Note: Continuous variables are presented as means ± standard deviation. * Significant effect *p* < 0.05. BMI: body mass index; ASMI: appendicular skeletal muscle mass index; ASM: appendicular skeletal muscle mass; LBM: lean body mass; Wt: weight; Ht: height; IPAQ: international physical activity questionnaire; ALT(GPT): alanine aminotransferase(glutamic pyruvate transaminase); AST(GOT): aspartate aminotransferase (glutamic oxalacetic transaminase); 25(OH)D: 25-hydroxyvitamin D; CBC: complete blood count; WBC: white bold cell count; RBC: red blood count; Hb: hemoglobin; HCT: hematocrit; MCV: mean corpuscular; MCH: mean corpuscular hemoglobin; MCHC: mean cell hemoglobin concentration; RDW: red cell distribution with; PLT: platelet count; MPV: mean platelet volume.

**Table 2 nutrients-12-01816-t002:** Changes of dietary intake during the 12-week study.

	Mean (S.E)		Time	Group	Time × Group
	Control (*n* = 57)	Intervention (*n* = 54)			
Energy (kcal)			0.145	0.346	0.713
Baseline	1486.0 (41.3)	1568.4 (42.4)			
12 weeks	1584.5 (71.7)	1626.7 (73.7)			
Carbohydrate (g)			0.603	0.173	0.663
Baseline	255.6 (6.8)	235.9 (7.0)			
12 weeks	226.2 (11.0)	243.9 (11.3)			
Protein (g)			0.014	0.892	0.854
Baseline	56.5 (2.0)	57.3 (2.1)			
12 weeks	63.7 (3.3)	63.6 (3.4)			
Fat (g)			0.125	0.743	0.202
Baseline	38.4 (2.1)	40.9 (2.2)			
12 weeks	45.7 (3.0)	41.5 (3.0)			
Vitamin A (ug RAE)			0.904	0.839	0.748
baseline	441.99 (44.12)	419.48 (45.34)			
12 weeks	422.94 (39.74)	428.19 (40.84)			
Vitamin D (ug)			0.178	0.593	0.047 *
Baseline	2.467 (0.40)	3.260 (0.42)			
12 weeks	4.317 (0.67)	2.904 (0.68)			
Vitamin E			0.606	0.947	0.998
Baseline	14.98 (0.98)	15.06 (1.01)			
12 weeks	15.53 (1.08)	15.60 (1.11)			
Vitamin K			0.010	0.884	0.824
Baseline	158.78 (25.59)	161.81 (26.30)			
12 weeks	261.71 (42.36)	248.55 (43.52)			
Thiamin (mg)			0.036	0.600	0.785
Baseline	1.891 (0.88)	2.582 (0.90)			
12 weeks	2.066 (0.88)	2.717 (0.90)			
Riboflavin (mg)			0.031	0.047	0.017 *
Baseline	1.208 (0.06)	1.236 (0.06)			
12 weeks	1.552 (0.08)	1.218 (0.08)			
Niacin (mg NE)			0.032	0.627	0.615
Baseline	9.913 (0.51)	10.584 (0.53)			
12 weeks	11.608 (0.78)	11.639 (0.81)			
Vitamin B_6_ (mg)			0.028	0.621	0.940
Baseline	1.671 (0.28)	1.892 (0.29)			
12 weeks	2.417 (0.41)	2.590 (0.42)			
Calcium (mg)			0.300	0.415	0.097
Baseline	435.68 (31.58)	459.52 (32.46)			
12 weeks	250.04 (32.82)	439.90 (33.72)			
Isoleucine (mg)			<0.001	0.722	0.511
Baseline	1392.44 (101.34)	1431.09 (104.16)			
12 weeks	2062.15 (167.93)	1920.21 (172.55)			
Leucine (mg)			0.002	0.673	0.385
Baseline	2583.17 (178.63)	2653.99 (183.58)			
12 weeks	3308.59 (196.48)	3066.50 (201.91)			
Lysine (mg)			0.704	<0.001	0.236
Baseline	1849.85 (162.66)	1984.54 (167.16)			
12 weeks	2709.38 (186.56)	2434.69 (191.71)			
Valine (mg)			<0.001	0.652	0.237
Baseline	1636.59 (108.60)	1707.20 (111.61)			
12 weeks	2176.60 (124.38)	1989.61 (127.82)			

Note: Mixed effect Model Repeated Measurement (MMRM) analysis was adjusted for age. * Significant main effect or interaction, *p* < 0.05. The data is based on a 24-hour recall of baseline and a food diary after 12 weeks, analysis was performed by a web-based computer nutrition analysis program (CAN-Pro. ver. 5.0, Korean Nutrition Society, 2015).

**Table 3 nutrients-12-01816-t003:** Muscle health and physical performance changes during the 12-week study period.

	Mean (S.E)		Time	Group	Time × Group
	Control	Intervention			
BMI (kg/m^2^)			<0.001	0.463	0.754
Baseline	23.61 (0.25)	23.88 (0.26)			
12 weeks	23.84 (0.25)	24.07 (0.26)			
Weight (kg)			<0.001	0.529	0.548
Baseline	60.81 (1.01)	61.74 (1.04)			
12 weeks	61.44 (1.03)	62.23 (1.05)			
Height (cm)			0.201	0.939	0.201
Baseline	160.40 (0.90)	160.51 (0.92)			
12 weeks	160.42 (0.90)	160.51 (0.92)			
Muscle health					
ASM (kg)			0.001	0.575	0.923
Baseline	16.77 (0.47)	17.16 (0.49)			
12 weeks	16.93 (0.47)	17.32 (0.48)			
ASMI (kg/m^2^)			0.032	0.534	0.818
Baseline	6.48 (0.12)	6.58 (0.12)			
12 weeks	6.53 (0.12)	6.65 (0.12)			
ASM/Wt (kg/kg, %)			0.764	0.801	0.465
Baseline	27.47 (0.45)	27.58 (0.45)			
12 weeks	27.44 (0.44)	27.65 (0.45)			
ASM/BMI [kg/(kg/m^2^)]			0.694	0.790	0.554
Baseline	0.71 (0.01)	0.71 (0.01)			
12 weeks	0.71 (0.01)	0.72 (0.01)			
LBM (kg)			0.008	0.697	0.027 *
Baseline	38.93 (0.92)	39.23 (0.95)			
12 weeks	38.97 (0.91)	39.70 (0.93)			
LBM/Ht (kg/m^2^)			0.005	0.641	0.019 *
Baseline	15.04 (0.21)	15.06 (0.21)			
12 weeks	15.09 (0.22)	15.29 (0.21)			
LBM/Wt (kg/kg, %)			0.295	0. 974	<0.001 *
Baseline	63.85 (0.82)	63.38 (0.85)			
12 weeks	63.29 (0.81)	63.68 (0.83)			
LBM/BMI [kg/(kg/m^2^)]			0. 273	0. 946	0.001 *
Baseline	1.65 (0.03)	1.64 (0.03)			
12 weeks	1.63 (0.03)	1.65 (0.03)			
Arm circumference (cm)			<0.001	0.108	0.542
Baseline	29.83 (0.26)	30.45 (0.27)			
12 weeks	30.15 (0.26)	30.82 (0.17)			
Calf circumference (cm)			0.039	0.366	0.936
Baseline	34.79 (0.35)	35.24 (0.36)			
12 weeks	34.33 (0.36)	34.76 (0.37)			
Femoral muscle strength (N)			0.813	0.397	0.839
Baseline	173.91 (6.89)	182.35 (7.08)			
12 weeks	172.39 (5.70)	182.70 (5.86)			
Femoral muscle strength/Wt (N/kg)			0.833	0.421	0.929
Baseline	2.85 (0.11)	2.98 (0.11)			
12 weeks	2.83 (0.09)	2.95 (0.09)			
Grip strength (kg)			0.134	0.096	0.799
Baseline	26.53 (1.02)	28.99 (1.04)			
12 weeks	25.89 (1.02)	28.51 (1.05)			
physical performance					
SPPB (score)			0.003	0.423	0.130
Baseline	11.52 (0.10)	11.64 (0.10)			
12 weeks	11.85 (0.07)	11.73 (0.07)			

Note: Mixed effect Model Repeated Measurement (MMRM) analysis was adjusted for age. * Significant main effect or interaction, *p <* 0.05. BMI: body mass index; ASMI: appendicular skeletal muscle mass index; ASM: appendicular skeletal muscle mass; LBM: lean body mass; Wt: weight, Ht: height; SPPB: short physical performance battery.

**Table 4 nutrients-12-01816-t004:** Blood measurement changes during the 12-week study period.

	Reference	Mean (S.E)		Time	Group	Time × Group
		Control	Intervention			
Glucose (mg/dL)				0.151	0.169	0.751
Baseline	74–106	97.59 (1.06)	99.71 (1.09)			
12 weeks		98.89 (1.09)	100.60 (1.12)			
Insulin (uIU/mL)				0.004	0.125	0.019 *
Baseline	1.1–11.6	6.54 (0.43)	7.51 (0.44)			
12 weeks		7.97 (0.48)	7.30 (0.49)			
Albumin (g/dL)				0.014	0.466	0.980
Baseline	3.5–5.2	4.60 (0.03)	4.56 (0.03)			
12 weeks		4.52 (0.03)	4.49 (0.03)			
ALT(GPT) (U/L)				0.247	0.415	0.461
Baseline	5–33	19.19 (1.24)	20.67 (1.28)			
12 weeks		21.04 (1.91)	24.18 (1.96)			
AST(GOT) (U/L)				0.265	0.759	0.803
Baseline	5–32	22.97 (0.95)	23.39 (0.98)			
12 weeks		24.13 (1.31)	24.93 (1.35)			
Creatinine (mg/dL)				0.213	0.665	0.332
Baseline	0.50–0.90	0.76 (0.02)	0.77 (0.02)			
12 weeks		0.75 (0.02)	0.78 (0.02)			
25(OH)D (ng/mL)				<0.001	0.543	<0.001 *****
Baseline	Deficiency: <20.0 ng/mL Insufficiency: 20.0~30.0 ng/mL Sufficiency: 30.1–100.0 ng/mL	35.83 (1.48)	34.53 (1.52)			
12 weeks		29.09 (1.35)	40.62 (1.38)			
CBC						
WBC (×10^3^/μL)				0.691	0.079	0.173
Baseline	3.4–10.6	5.07 (0.15)	5.32 (0.16)			
12 weeks		4.98 (0.16)	5.49 (0.16)			
RBC(×10^6^/μL)				0.042	0.077	0.331
Baseline	3.58–4.91	4.33 (0.04)	4.42 (0.04)			
12 weeks		4.35 (0.04)	4.49 (0.05)			
Hb(g/dL)				0.204	0.105	0.331
Baseline	10.7–15.3	13.43 (0.15)	13.76 (0.16)			
12 weeks		13.45 (0.17)	13.89 (0.17)			
HCT (%)				0.024	0.076	0.224
Baseline	32.4–44.9	40.37 (0.43)	41.29 (0.44)			
12 weeks		40.57 (0.49)	41.97 (0.50)			
MCV (fL)				0.492	0.747	0.151
Baseline	80.0–99.0	93.31 (0.61)	93.44 (0.62)			
12 weeks		93.08 (0.63)	93.52 (0.65)			
MCH (pg)				0.834	0.665	0.269
Baseline	26.2–33.4	31.08 (0.42)	30.61 (0.43)			
12 weeks		30.85 (0.25)	30.94 (0.25)			
MCHC (g/dL)	31.5-35.0			0.004	0.835	0.982
Baseline		33.27(0.09)	33.24(0.09)			
12 weeks		33.11(0.08)	33.09(0.08)			
RDW (%)				0.025	0.462	0.951
Baseline	12.0–15.7	13.33 (0.12)	13.19 (0.13)			
12 weeks		13.24 (0.14)	13.09 (0.15)			
PLT (×10^3^/μL)				0.002	0.922	0.370
Baseline	134–387	221.81 (5.91)	218.65 (6.08)			
12 weeks		227.63 (6.28)	229.15 (6.46)			
MPV (fL)				0.021	0.435	0.162
Baseline	6.0–9.9	7.75 (0.09)	7.81 (0.09)			
12 weeks		7.63 (0.09)	7.78 (0.10)			

Note: Mixed effect Model Repeated Measurement (MMRM) analysis was adjusted for age. * Significant main effect or interaction, *p <* 0.05. Reference: Ajou University Korea standard was used as the reference value; ALT(GPT): alanine aminotransferase(glutamic pyruvate transaminase); AST(GOT): aspartate aminotransferase(glutamic oxaiacetic transaminase); 25(OH)D: 25 hydroxyvitamin D; CBC: complete blood count; WBC: white bold cell count; RBC: red blood count; Hb: hemoglobin; HCT: hematocrit; MCV: mean corpuscular; MCH: mean corpuscular hemoglobin; MCHC: mean cell hemoglobin concentration; RDW: red cell distribution with; PLT: platelet count; MPV: mean platelet volume.

**Table 5 nutrients-12-01816-t005:** Muscle health during the 12-week study period in participants aged 50–64 years (n = 90).

	Mean (S.E)		Time	Group	Time × Group
	Control	Intervention			
ASM (kg)			0.007	0.380	0.654
Baseline	16.74 (0.50)	17.37 (0.56)			
12 weeks	16.88 (0.49)	17.56 (0.55)			
ASMI (kg/m^2^)			0.094	0.424	0.543
Baseline	6.43 (0.13)	6.59 (0.14)			
12 weeks	6.48 (0.13)	6.66 (0.14)			
ASM/Wt (kg/kg, %)			0.602	0.553	0.177
Baseline	27.57 (0.47)	27.88 (0.53)			
12 weeks	27.50 (0.47)	28.03 (0.53)			
ASM/BMI [kg/(kg/m^2^)]			0.539	0.524	0.214
Baseline	0.71 (0.01)	0.73 (0.02)			
12 weeks	0.71 (0.01)	0.73 (0.02)			
LBM (kg)			0.047	0.444	0.049 *
Baseline	38.83 (0.96)	39.70 (1.08)			
12 weeks	38.83 (0.94)	40.15 (1.05)			
LBM/Ht (kg/m^2^)			0.031	0.473	0.033 *
Baseline	14.94 (0.22)	15.09 (0.25)			
12 weeks	14.94 (0.22)	15.27 (0.24)			
LBM/Wt (kg/kg, %)			0.313	0.749	<0.001 *
Baseline	64.09 (0.90)	64.03 (1.01)			
12 weeks	63.45 (0.88)	64.37 (0.98)			
LBM/BMI [kg/(kg/m^2^)]			0.333	0.630	0.001 *
Baseline	1.66 (0.03)	1.68 (0.04)			
12 weeks	1.65 (0.03)	1.69 (0.04)			
Femoral muscle strength (N)			0.820	0.357	0.864
Baseline	171.79 (7.64)	182.41 (8.55)			
12 weeks	170.18 (6.15)	182.61 (6.88)			
Femoral muscle strength/Wt (N/kg)			0.832	0.438	0.952
Baseline	2.83 (0.12)	2.97 (0.13)			
12 weeks	2.81 (0.10)	2.94 (0.11)			

Note: Mixed effect Model Repeated Measurement (MMRM) analysis was adjusted for age.* Significant main effect or interaction, *p <* 0.05. ASMI: appendicular skeletal muscle mass index; ASM: appendicular skeletal muscle mass; LBM: lean body mass; Wt: weight; Ht: weight.

**Table 6 nutrients-12-01816-t006:** Muscle health during the 12-week study period in participants aged 65 years and older (n = 21).

	Mean (S.E)		Time	Group	Time × Group
	Control	Intervention			
ASM (kg)			0.054	0.389	0.326
Baseline	17.58 (1.29)	16.25 (0.90)			
12 weeks	17.90 (1.36)	16.36 (0.96)			
ASMI (kg/m^2^)			0.086	0.616	0.330
Baseline	6.79 (0.34)	6.58 (0.24)			
12 weeks	6.90 (0.35)	6.61 (0.25)			
ASM/Wt (kg/kg, %)			0.826	0.370	0.330
Baseline	27.43 (1.08)	26.37 (0.76)			
12 weeks	27.67 (1.13)	26.23 (0.79)			
ASM/BMI [kg/(kg/m^2^)]			0.795	0.277	0.275
Baseline	0.70 (0.04)	0.65 (0.03)			
12 weeks	0.71 (0.04)	0.64 (0.03)			
LBM (kg)			0.031	0.317	0.603
Baseline	40.70 (2.53)	37.32 (1.78)			
12 weeks	41.04 (2.63)	37.86 (1.85)			
LBM/Ht (kg/m^2^)			0.030	0.506	0.611
Baseline	15.70 (0.60)	15.15 (0.42)			
12 weeks	15.84 (0.64)	15.36 (0.44)			
LBM/Wt (kg/kg, %)			0. 812	0. 301	0.725
Baseline	63.49 (1.90)	60.85 (1.33)			
12 weeks	63.46 (1.98)	61.02 (1.39)			
LBM/BMI [kg/(kg/m^2^)]			0.947	0.233	0.772
Baseline	1.63 (0.08)	1.50 (0.06)			
12 weeks	1.63 (0.08)	1.50 (0.06)			
Femoral muscle strength (N)			0.958	0.989	0.936
Baseline	184.68 (16.71)	184.38 (11.75)			
12 weeks	183.80 (15.72)	185.14 (11.05)			
Femoral muscle strength/Wt (N/kg)			0.983	0.665	0.926
Baseline	2.90 (0.26)	3.05 (0.19)			
12 weeks	2.90 (0.23)	3.01 (0.16)			

Note: Mixed effect Model Repeated Measurement (MMRM) analysis was adjusted for age. ASMI: appendicular skeletal muscle mass index; ASM: appendicular skeletal muscle mass; LBM: lean body mass; Wt: weight; Ht: height.

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
