# Peer review of "Leucine-Enriched Protein Supplementation Increases Lean Body Mass in Healthy Korean Adults Aged 50 Years and Older: A Randomized, Double-Blind, Placebo-Controlled Trial"

_nutrients, 2020, doi:10.3390/nu12061816_

Round 1

Reviewer 1 Report

This paper is well written and easy to read, however there are several comments.

(1) There are clear differences of LBM in between control and leucine supplemental group, but there is no significant differences of ASMI. You should add the comments of this difference in Discussion.

(2) In the current experiment, women participants (approximately number is 40) are enough to evaluate as subgroup analysis.

(3) In the table 1, the order of column should be control, intervention and P, because other tables are this order.

(4) There is clear increase of serum 25(OH)D in intervention group. Therefore, it is interesting to see correlation between 25(OH)D change(12w - base) and LBM/Wt x 10 change (12w - base) in individual control and intervention group, and also in combination of control and intervention groups. If you find new interesting data, you should add the importance of vitamin D supplement for muscle mass in the Discussion.

Author Response

We greatly appreciate the opinions of the reviewers. All comments have been an important guide for improving research. We were able to improve the quality of this manuscript by respecting the opinions of reviewers. Please see the attachment.

We did our best to improve the quality of the manuscript. We believe that the results of this study are relevant to the scope of the journal and will attract readers' attention. Thank you for your consideration.

Sincere you

Reviewer 2 Report

Dear authors,

This study sought to investigate the effects of leucine-enriched protein supplementation on muscle condition in adults between 50 and 80 years of age. This double-blind randomized study was well planned and conducted. There was a focus on the younger half of the study cohort (50-64 years), where the effects of a protein supplement to prevent sarcopenia is lacking. The manuscript is generally well written, however, the language needs to be improved and major and minor comments to the authors are included below.

Major comments

  • A hypothesis is lacking in the current study and should be included.
  • Results: 3.3 Dietary intake and Table 2. Is the nutritional value of the supplements accounted for in the dietary intake? If not, they should be as they contribute with substantial amounts of protein (and leucine), carbohydrates, vitamin D and calcium. This is also discussed in line 340 which should be changed accordingly. Also, in table 2 it says nothing on whether the dietary data describes the participants daily intake, or if it is based on the food diary or 24-hour recall.
  • The intake of protein and leucine from the supplement is emphasized in the title and in the introduction, however vitamin D seems to also be of importance in the discussion. Therefore, the role of vitamin D in the current context should be explained better in the introduction and possibly be mentioned in the title.

Minor comments

  • Line 34: Is aging in general a “public health issue”. Should be rewritten.
  • Line 107: Was the supplements mixed in water or milk? Important distinction. Should be specified in methods and discussed in limitations. And was this milk included in estimated dietary intake?
  • Lines 102-105: Specify how frequent the supplement is administered (daily?).
  • Line 110: Were the participants instructed to do some form of exercise? Why did they receive a resistance-exercise program brochure?
  • Lines 252-253: Must specify groups.
  • Lines 313-314: What was the intervention in the study described? And has this hypothesis been stated before?
  • Line 344: A new hypothesis should not be presented here.
  • Line 346: 1-5% loss of muscle mass per day sound like a lot. Can this be accurate?
  • Line 369: Is it not well known that there are sex differences regarding protein metabolism? And why is protein metabolism compared with lipid and carbohydrate metabolism?
  • Tables 2-5 needs a legend describing the statistics used and explanation of the different p-values.
  • Vitamin D intake is described using both IU and ug (and also g in line 198). Vitamin D circulating levels is described using both uIU/ml and ng/ml. Should use only one unit or show conversion in brackets, e.g.: 800 IU (20 ug).
  • Table 4: Vitamin D: Need to add a source describing that 30 ng/ml circulating 25(OH)D is “enough”. Should use standardized terms to describe the different concentrations, such as: deficiency, insufficiency, sufficiency.
  • Conclusion: Is this the only study to include pre-elderly subjects for a supplementation study? Be more specific or remove from the conclusion.

Author Response

(The authors gave the same response as above.)
